# Ozone inactivation of airborne influenza and lack of resistance of respiratory syncytial virus to aerosolization and sampling processes

**Marie-Eve Dubuis**[1,2], **Étienne Racine**[3], **Jonathan M. Vyskocil**[1,2], **Nathalie Turgeon**[1], **Christophe Tremblay**[2], **Espérance Mukawera**[4,5], **Guy Boivin**[6,7], **Nathalie Grandvaux**[4,5], **Caroline Duchaine**[1,2]*

**1** Centre de Recherche de l'Institut Universitaire de Cardiologie et de Pneumologie de Québec–Université Laval, Quebec City, Quebec, Canada, **2** Département de Biochimie, de Microbiologie et de Bio-informatique, Faculté des Sciences et de Génie, Université Laval, Quebec City, Quebec, Canada, **3** Faculté de Médecine, Université Laval, Quebec City, Quebec, Canada, **4** Centre de Recherche du Centre Hospitalier de l'Université de Montréal, Montreal, Quebec, Canada, **5** Département de Biochimie et Médecine Moléculaire, Faculté de Médecine, Université de Montréal, Montreal, Quebec, Canada, **6** Centre de Recherche du Centre Hospitalier Universitaire de Québec–Université Laval, Quebec City, Quebec, Canada, **7** Département de Microbiologie-Infectiologie et d'Immunologie, Faculté de Médecine, Université Laval, Quebec City, Quebec, Canada

* Caroline.Duchaine@bcm.ulaval.ca

**Data Availability Statement:** All relevant data are within the paper.

## Abstract

Influenza and RSV are human viruses responsible for outbreaks in hospitals, long-term care facilities and nursing homes. The present study assessed an air treatment using ozone at two relative humidity conditions (RHs) in order to reduce the infectivity of airborne influenza. Bovine pulmonary surfactant (BPS) and synthetic tracheal mucus (STM) were used as aerosols protectants to better reflect the human aerosol composition. Residual ozone concentration inside the aerosol chamber was also measured. RSV's sensitivity resulted in testing its resistance to aerosolization and sampling processes instead of ozone exposure. The results showed that without supplement and with STM, a reduction in influenza A infectivity of four orders of magnitude was obtained with an exposure to 1.70 ± 0.19 ppm of ozone at 76% RH for 80 min. Consequently, ozone could be considered as a virucidal disinfectant for airborne influenza A. RSV did not withstand the aerosolization and sampling processes required for the use of the experimental setup. Therefore, ozone exposure could not be performed for this virus. Nonetheless, this study provides great insight for the efficacy of ozone as an air treatment for the control of nosocomial influenza A outbreaks.

## Introduction

Viral outbreaks, including those caused by SARS-CoV-2, influenza and respiratory syncytial virus (RSV), are a major concern for hospitals, long-term care facilities and nursing homes [1–3]. Seasonal outbreaks of influenza occur annually and the threat of a pandemic is always present [4, 5]. The risk of developing complications from an influenza infection is greater for young children, pregnant women, adults that are 65 years and older, and people suffering

**Funding:** CD: Natural Sciences and Engineering Research Council of Canada (NSERC) Discovery Grant (RGPIN-2020-04284). https://www.nserc-crsng.gc.ca CD and NG: Quebec Respiratory Health Network team grant. https://rsr-qc.ca MED and CT: Natural Sciences and Engineering Research Council of Canada MED: Institut de Recherche Robert-Sauvé en Santé et en Sécurité du Travail. irsst.qc.ca The funders had no role in study design, data collection and analysis, decision to publish, or preparation of the manuscript.

**Competing interests:** The authors have declared that no competing interests exist.

from chronic diseases or immunosuppression [6]. Moreover, RSV is of great concern because almost every child is infected with this virus by the time they are 2 years old [7]. Adults can also be infected by RSV [8], particularly elderly individuals and those with underlying health problems, such as immunosuppression and cardiopulmonary diseases [9–12]. Complications from RSV such as bronchiolitis and pneumonia are common [7, 13] and outbreaks can occur in hospital settings, pediatric care units and neonatal intensive care units [14–17].

Previous studies have shown that airborne influenza genomes are present in hospital settings [18, 19], sometimes at concentrations of up to $10^5$ copies/m$^3$ (unpublished data from authors). RSV genomes have been detected in the air of an emergency clinic [20], in hospital rooms with RSV-infected patients [21] and in a pediatric acute care ward [22]. Only a small number of positive samples (2.3%) were collected in the air of the pediatric acute care ward, which suggests that airborne RSV transmission is not likely [22]. On the other hand, another study detected airborne infectious RSV in the air around infants that were hospitalized with bronchiolitis in a pediatric ward and an intensive care. Mean concentrations were 3.71 X $10^5$ PFU/m$^3$ in the pediatric ward and 4.09 X $10^5$ PFU/m$^3$ in the intensive care unit [23].

Because of the significant risks associated with influenza and RSV, a means to control the potential aerosol transmission of these diseases would be beneficial for health-care facilities. For this purpose, we have tested an air disinfection protocol using ozone. The experimental setup has previously been used for phages and murine norovirus [24]. The results that were obtained were promising, even when using lower concentrations of ozone than those used by Tseng and Li (2006) for the inactivation of airborne phages [25].

According to Vejerano and Marr (2018), a single human-produced evaporating respiratory droplet of 60 μm (roughly 0.1 nl) is estimated to compose 1 ng of salt, 1 ng of protein and 0.06 ng of surfactant. They have also suggested that the surfactant of the droplet may protect the lipid membrane of viruses [26]. Additionally, evidence suggests that RSV is protected when sucrose is added to the media [27–30].

The aim of this study was to assess the virulence of two enveloped viruses, influenza A and RSV, after being aerosolized in the laboratory and exposed to an air disinfection protocol using ozone at two relative humidity conditions (RHs) in a rotating aerosol chamber. Bovine pulmonary surfactant (BPS) and synthetic tracheal mucus (STM) were added to influenza A aerosols in order to test any protective effects they may have on the virus. Due to the sensitivity of RSV to various testing conditions, only the preliminary experiments were conducted for this virus. These experiments tested the ability of RSV to withstand the aerosolization and sampling processes.

## Materials and methods

### Influenza A virus and host cells

Influenza A/Michigan/45/2015 (clinical strain A/Quebec/22578/2016 provided by Guy Boivin) was propagated in MDCK (CCL-34, ATCC) and ST6GalI-MDCK cells. Host cells were cultivated in Minimum Essential Medium (MEM; Gibco, Grand Island, USA) with 10% Fetal Bovine Serum (FBS; Wisent, Saint-Jean-Baptiste, CANADA). Two viral lysate stocks were prepared: the first contained 9.22 X $10^7$ TCID$_{50}$/ml and the second contained 1.68 X $10^7$ TCID$_{50}$/ml. Viral lysate subsamples were stored at -80°C in volumes of 30 ml or 27 ml.

### Respiratory syncytial virus and host cells

The RSV-A2 strain (10-247-000; Advanced Biotechnologies Inc.) was amplified in HEp-2 cells (CCL-23; ATCC), purified on a 30% sucrose cushion [31, 32], and resuspended in 50mM Hepes and 5% sucrose in Opti-MEM (Gibco Grand Island, USA). Aliquots of 150 μl were

stored at -80˚C until use. The stock was titrated in 6-well plates at a concentration of 2.52 X $10^8$ PFU/ml. For all other experiments, the titration was performed using the $TCID_{50}$ technique, as described by Sun and López [33].

## Influenza A aerosolization, exposure to ozone and sampling

One viral lysate subsample was thawed prior to each nebulization. Ten microliters of Antifoam A concentrate (Sigma-Aldrich, St-Louis, USA) was added to the nebulizing liquid to prevent foaming. Based on the estimated composition of a human droplet, we selected two substances to supplement the nebulizing liquid: BPS (lipid-based) and STM (protein-based). In order to compare the effectiveness of each supplement at protecting the airborne virus, they were each tested separately and added to the nebulizing solution at a final concentration of 10%. This method was used by Kormuth et al. (2018) with extracellular material produced by human bronchial epithelial cells (HBE ECM) [34]. However, we could not use pooled washes of HBE ECM because a volume of 60 ml of the supplement was needed for our experiments. Depending on the test conditions, the nebulizing liquid was either left as is (no treatment; NT) or supplemented with 10% of BPS (BPS treatment; BPST) (provided by BLES Biochemicals Inc.) or 10% of STM (STM treatment; STMT). The STM was composed of 0.3 g of albumin (05470-1G; Sigma, St-Louis, USA) and 1.2 g of pig gastric mucin-type III (M1778-10G, Sigma) in 28.2 ml of buffer solution (154 mM NaCl, 3 mM $CaCl_2$, 15 mM $NaH_2PO_4$/$Na_2HPO_4$; pH 7.4) [35].

Aerosolization was performed as described in [24]. Briefly, the virus stock was nebulized at 20 psi with a 6-jet Collison nebulizer (BGI, Waltham, USA). The Collision nebulizer was connected to compressed medical-grade air for 10 minutes inside a 55-L rotative environmental aerosol chamber [36], which was placed in a biosafety level (BSL) II cabinet. Diffusion dryer tubes of different lengths were used to adjust the RH inside the chamber. RH values were monitored with a probe (model STH-ID 300, KIMO Instruments, Montpon, FRANCE). Resulting RH values were 31.6 ± 3.9% and 80.6 ± 1.9% for the NT and BPST experiments. For the STMT experiments, RH values of 34.3 ± 3.6% and 75.8 ± 2.2% were obtained. The nebulizing step was followed by a 10-minute waiting time to ensure an even mix of aerosols in the chamber. Aerosol distribution size and number were assessed using an Aerodynamic Particle Sizer (APS) (model 3321; TSI Inc., Shoreview, USA). Particle dilution was necessary for accurate reads, and therefore a 1/100 dilutor with a 1/20 capillary (model 3302A, TSI Inc.) was used with the APS. The mass median aerodynamic diameters (MMAD) for the NT experiments were 1.22 ± 0.03 μm at 32% RH and 1.26 ± 0.03 μm at 81% RH. For the BPST, the MMADs were 1.18 ± 0.03 μm at 32% RH and 1.35 ± 0.02 μm at 81% RH, and the MMADs for the STMT were 1.27 ± 0.02 μm at 34% RH and 1.51 ± 0.07 μm at 76% RH.

Next, aerosols were exposed to air (designated as the reference conditions) or a mixed gas of approximately 21% of ozone in air for 0, 30 or 60 minutes at low or high RHs. Ozone was produced by a generator (model 201705004A210Y, EMO3) and injected into the chamber at a flowrate of 0.4 L/min for 30 seconds. Ozone injection continued until a concentration of 0.23 ppm ± 0.03 ppm was obtained for the NT and BPST experiments. The injection continued for 50 seconds for the STMT experiments, and a concentration of 1.70 ± 0.19 ppm was reached.

Samples were collected using a SKC BioSampler (SKC Inc., Eighty Four, USA) filled with 20 ml of MEM and connected to an SKC vacuum pump (model 228 ± 9605; SKC Inc.) for 20 minutes at a flowrate of 12.5 L/min. As mentioned in Dubuis et al. (2020), due to the 20-minute sampling step, a 10-minute exposure period was added to the tested exposure times of 0, 30 and 60 minutes. Consequently, the exposure times for the NT and BPST experiments were 10, 40 and 70 minutes.

For the STMT experiments, a new ozone probe (PortaSens III; Analytical Technology Inc., USA) coupled with two sensors (500–2000 ppb; 00–1163 and 1–5 ppm: 00–1008) was used. The sensors could be quickly switched depending on the ozone concentration during the experiments. The sensitivity of the sensors is considered to be 1% of their ranges, which are 20 ppb (500–2000 ppb) and 50 ppb (1–5 ppm), with a resolution of 1 ppb and 0.01 ppm, respectively. Because the probe was able to directly measure the ozone concentration inside the chamber, readings were performed every 10 minutes, which allowed us to monitor the residual ozone concentration over time. After the nebulizing step, ozone was injected into the chamber for 50 seconds. Following this step, even distribution of aerosols was obtained within a 10-minute waiting period. The ozone concentration was then measured, followed by an APS reading. Because of the 10-minute mixing of aerosols prior to the ozone and APS readings, a 10-minute period needed to be added to the exposure times, as well as a 10-minute period from the sampling step. Total exposure times for the STMT experiments were 20, 50 and 80 minutes.

As explained in previously published work, control samples were drawn from the nebulizing stock to monitor the viral concentrations throughout the experiments [24]. The effect of ozone in the SKC BioSampler was also assessed and considered negligible [24].

## RSV nebulizing and sampling assays

**Virus resistance in air sampler.** Solutions of DMEM + 5% sucrose and DMEM + 20% sucrose were prepared and stored at 4˚C until use. Final sucrose concentrations of 5% and 20% were obtained by following the methods from a previous report by Grosz et al. [30]. A conical tube was filled with 50 ml of the 5% sucrose solution and spiked with three aliquots of the frozen RSV stock. The solution was vortexed and divided between two SKC BioSamplers (20 ml each). An aliquot of the spiked virus solution (initial concentration) was stored on ice. The SKC BioSamplers were placed inside a BSL II cabinet and were not connected to an aerosol chamber. The BioSamplers sampled clean air from the BSL II cabinet for a period of 20 minutes. Aliquots for both SKC BioSamplers were extracted (final concentration) and kept on ice. $TCID_{50}$ was performed immediately for each aliquot. The same procedure was repeated with the spiked 20% sucrose solution.

**Virus resistance in two nebulizers.** An assay was conducted to determine whether the nebulizing process alone could inactivate RSV. The 6-jet Collison nebulizer and a gentler device, the Aeroneb Lab nebulizer (Aerogen Inc., Galway, Ireland) were used.

The 6-jet Collison nebulizer was filled with 20 ml of DMEM mixed with one aliquot of the frozen viral stock. An aliquot that represented the initial concentration inside the nebulizer was extracted and stored on ice. The 6-jet Collison nebulizer was connected to the same experimental setup as the one for influenza A aerosolization described above, and run for 10 minutes. A second aliquot was extracted and kept on ice, this one containing the final nebulizer concentration. $TCID_{50}$ was performed on these two aliquots. The 6-jet Collison experiment was performed in sextuplicate.

For the Aeroneb nebulizer, two different solutions were prepared: one with 5% sucrose and the other with 20%. The 5% sucrose solution was spiked with the frozen RSV stock (three aliquots of frozen RSV in 50 ml of solution) and 7 ml of this mixture was used to fill the nebulizer. An aliquot was immediately drawn and kept on ice (initial concentration). The Aeroneb was connected to a desiccant tube with a collection cup, which was then connected to the aerosol chamber setup. The collection cup was disinfected with 70% ethanol and placed in an ice bath. Air dilution was performed at a rate of 2 L/min to push the produced aerosols into the chamber and the nebulizer was run until empty (18 minutes). An aliquot from the

collection cup (final concentration) was kept on ice and immediately titrated using the TCID$_{50}$ method. This experiment was performed once, after which the 20% sucrose solution assay was performed using the same method.

## RNA extraction, RT-qPCR and influenza A titration

Three aliquots of each sample were kept at -80˚C: one for culture, one for RT-qPCR and one as a backup. Two aliquots from the nebulizing liquid were also frozen. To better conserve the virus, each aliquot was supplemented with bovine serum albumin factor V (BSA; Gibco, Canada) following the proportions described by Turgeon et al. (2019).

The QIAamp Viral RNA Kit (QIAGEN, Hilden, GERMANY) was used for RNA extraction without the RNA carrier. Elution was performed in two 40-μl volumes of TE buffer (80 μl total). One-step Real-Time PCR (RT-qPCR) was conducted with the Bio-Rad CFX384 thermocycler (Bio-Rad Laboratories, Mississauga, CANADA) for 5 μl of the extracted RNA. For the controls, 1/10 000 dilutions were required to fit the standard curve. The following conditions were used: 50˚C for 10 min, 94˚C for 3 min, 40 cycles at 94˚C for 15 s and 60˚C for 1 min. Each plate contained a standard curve of a 10-fold dilution series of influenza A plasmid preparation, as well as no template controls (NTC). Specific influenza A primers and a probe were used [37].

For infectious virus titration, TCID$_{50}$ was performed using MDCK or ST6GalI-MDCK cells. Plates were incubated at 37˚C with 5% $CO_2$ for 72h.

## Calculations and statistical analysis

Calculations were performed as described in Dubuis et al. (2020) [24]. Briefly, infectious ratios (IRs) for the reference conditions (air) and the ozone conditions were obtained by dividing mean culture counts (PFU/ml) with mean qPCR results. The normalized infectious ratios (NIRs) were then obtained by normalizing IRs with the 6-jet Collison nebulizer's initial viral concentrations. Relative infectious ratios (RIRs) were calculated using the ozone NIRs and the corresponding median air NIRs. Consequently, the aerosolization, humidity and aerosol-aging effects are not included in the RIR values.

Data were analyzed using a three-way ANOVA. The three fixed effects were treatments, temperature and humidity with three, three and two levels, respectively. Statistical models included two-by-two interaction terms. Univariate normality was verified by applying the Shapiro-Wilk tests on the error distribution from the statistical models. The Brown-Forsythe variation of Levene's test statistic was used to verify equal variances. A logarithm transformation was used for all statistical analyses, because normality and variance assumptions were not fulfilled. In order to perform comparisons between RH and exposure times, low RH values (32% and 34%) and high RH values (76% and 81%) were considered to be identical. Exposure times were combined into three groups: short (10 min and 20 min), medium (40 min and 50 min) and long (70 min and 80 min). These groups were then compared with each other. For these statistical analyses, results with P-values < 0.05 were considered significant. Analyses were performed using SAS version 9.4 (SAS Institute Inc., Cary, NC).

Ozone doses were estimated by calculating the area under concentration-vs-time curves, for RH values of 34% and 76%. Concentration-vs-time curves were fitted to observed data by regression analysis for exposure times between 10 min and 70 min (see S1 File for details). Regression analyses were performed in RStudio (version 1.2.5033). The areas under the curves were calculated numerically with a trapezoid scheme coded in C++ and validated against analytical computations. For exposure times between 0 and 10 min, extrapolation of the fitted concentration-vs-time curves was deemed too unreliable for dose calculations.

Accumulated dose between 0 and 10 min was instead estimated by assuming that ozone concentration was constant over that time interval and equal to the fitted concentration value at 10 min. Consequently, calculated ozone doses represent floor values. Confidence intervals were not reported for dose calculations because the uncertainty associated to the constant concentration assumption was much larger than the width of the confidence bands around the regressions curves.

## Results

### Ozone concentrations and doses

The ozone probe was used to monitor the concentration of ozone inside the aerosol chamber throughout the exposure periods. The initial readings were performed 10 min after ozone injection and subsequent readings were conducted at 10-min intervals. Therefore, the ozone concentrations measured with the PortaSens III probe do not include the additional 10-min sampling period. Although ozone concentrations between the two RHs were at times highly varied, the same quantity of ozone was injected for both RHs. The injected ozone concentration for all experiments was 1.70 ± 0.19 ppm, which was determined based on the ozone measurements at 34% RH after the 10-min mixing period. As shown in Fig 1, the ozone concentration decreased gradually over time at 34% RH. At 76% RH, the initial concentration was much lower, at 0.44 ± 0.12 ppm, and it decreased to close to zero after 30 min.

The calculated ozone doses floor values for both RH levels at 10-minute intervals are presented in Table 1. Even if the ozone injection was identical, after an exposure time of 70 min, the ozone dose floor value at 34% was much higher, at 58.67 ppm · min, than the dose at 76%, at 7.41 ppm · min. Furthermore, the values increase by a factor of two between 10 and 70 minutes at 76%, while they increase by a factor of three at 34% RH.

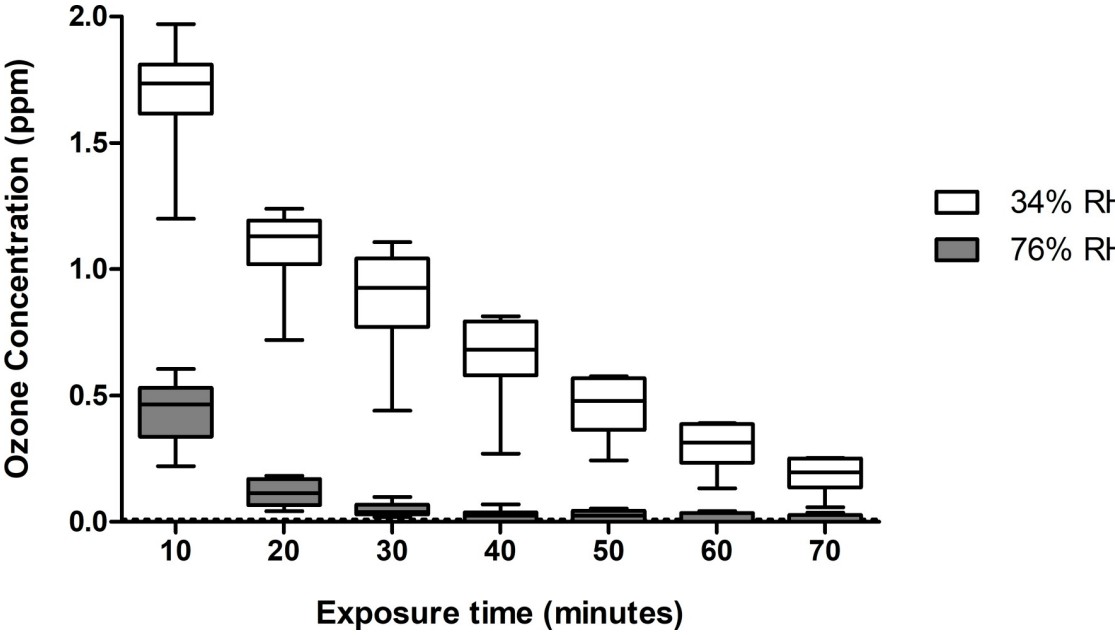

**Fig 1. Ozone concentrations measured at 34% and 76% RH at 10-min intervals over a maximum exposure time of 70 min.** The boxes at 10 minutes of exposure represent a total of 18 readings for both RHs. Boxes at 20, 30 and 40 min represent twelve readings each and boxes at 50, 60 and 70 min represent six readings each.

**Table 1. Ozone doses floor value (ppm · min) for each exposure time at 34% and 76% RH.**

| Exposure Time (min) | Ozone dose (ppm · min) | |
|---|---|---|
| | 34% RH | 76% RH |
| 10 | 16.90 | 3.69 |
| 20 | 31.11 | 5.72 |
| 30 | 41.05 | 6.49 |
| 40 | 48.01 | 6.89 |
| 50 | 52.88 | 7.13 |
| 60 | 56.29 | 7.29 |
| 70 | 58.67 | 7.41 |

## Reference conditions for influenza A

Fig 2 shows the NIRs for influenza A for the NT (Fig 2A), BPST (Fig 2B), and STMT (Fig 2C) experiments. As explained in Dubuis et al. (2020), normalized infectious ratio (NIR) values close to one mean that the reference conditions (exposure to air) do not cause a decrease in infectivity compared to the nebulizer content. NIR values below one indicate a reduction in infectivity for aerosols exposed to the reference conditions [24]. It is worth mentioning that the nebulized concentrations of infectious viruses for the NT, BPST and STMT experiments were $6.96 \times 10^7 \pm 9.40 \times 10^7$ PFU/ml, $1.58 \times 10^7 \pm 5.35 \times 10^6$ PFU/ml and $1.58 \times 10^7 \pm 4.54 \times 10^6$ PFU/ml, respectively. The concentrations of sampled viruses were lower, at $1.58 \times 10^5 \pm 1.92 \times 10^5$ PFU/ml for the NT, $2.20 \times 10^4 \pm 3.49 \times 10^4$ for the BPST and $1.58 \times 10^4 \pm 2.26 \times 10^4$ for the STMT.

Interactions with exposure times did not affect the NIR (p = 0.93) for both supplement and non-supplement conditions, meaning that the NIR values remained stable throughout the experiments. Additionally, the interactions between RH and both the NT (p = 0.74) and the STMT (p = 0.41) did not affect the NIRs. Indeed, the infectivity loss was equal to or less than one order of magnitude for both RHs. However, the interaction between BPST and RH (p < 0.01) was significant. When BPS was added to the nebulizer solution, there was an immediate decrease in infectivity of one to two orders of magnitude when RH was at 32%, and a decrease of three to four orders of magnitude when RH was at 81%. These NIR values reveal differences between the BPST and the NT (p < 0.01) and the BPST and the STMT (p < 0.01). The BPST significantly reduced the infectivity of Influenza A compared to the NT or

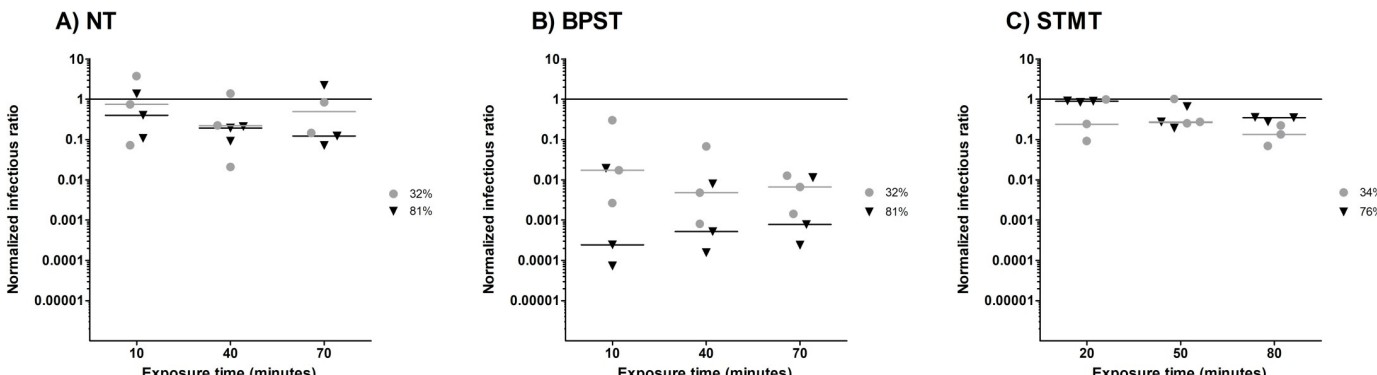

**Fig 2. Normalized Infectious Ratios (NIRs) for influenza A at three exposure times and two Relative Humidity (RH) levels.** Grey circles (●) represent low RH values (32% or 34%) and black triangles (▼) represent high RH values (76% or 81%). A) Without supplement (NT), B) with 10% bovine pulmonary surfactant (BPST) and C) with 10% synthetic tracheal mucus (STMT). The nebulizer content is represented by the solid line.

compared to the STMT. Therefore, the NT and STMT conditions are more appropriate than the BPST for the conservation of influenza's infectivity throughout the reference condition.

### Relative infectious ratios

The relative infectious ratios (RIRs) are presented in Fig 3, which represents the only effects that were observed as a result of ozone exposure. There was no significant difference (p = 0.31) between the NT (Fig 3A) and the BPST (Fig 3B). This means that the BPS did not protect the remaining infectious airborne influenza A from an exposure to 0.23 ppm ± 0.03 ppm of ozone. Significant differences were observed between interactions at 81% RH (p = 0.02) and at exposure times of 40 min (p < 0.01) and 70 min (p < 0.01), during which RIRs decreased by one to two orders of magnitude.

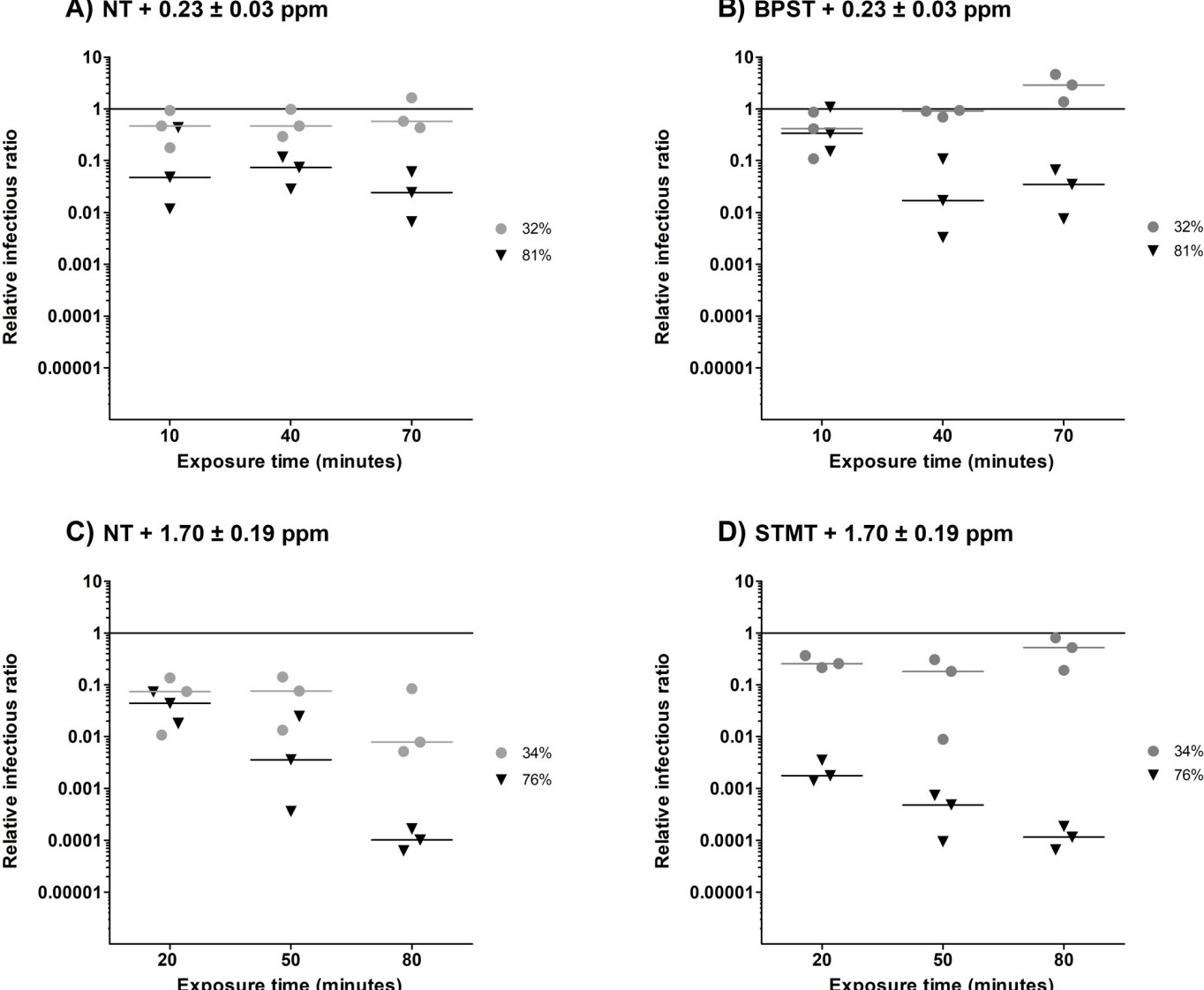

**Fig 3. The effect of ozone on influenza A infectivity at two Relative Humidity (RH) levels and three exposure times.** The reference value (Fig 2) is represented by a solid line. Grey circles (●) represent low RH values (32 or 34% RH) and black triangles (▼) represent high RH values (76 or 81%). A) No treatment (NT) and exposed to 0.23 ± 0.03 ppm of ozone, B) 10% bovine pulmonary surfactant (BPST) and exposed to 0.23 ± 0.03 ppm of ozone, C) NT and exposed to 1.70 ± 0.19 ppm of ozone, and D) 10% synthetic tracheal mucus (STMT) and exposed to 1.70 ± 0.19 ppm of ozone.

RIR results for the NT samples (Fig 3A and 3C) exhibited strong differences between those with ozone exposures of 0.23 ppm ± 0.03 ppm and those with exposures of 1.70 ± 0.19 ppm ($p < 0.01$). The maximum reduction in infectivity was four orders of magnitude, which was obtained after 80 min at 76% RH for an ozone exposure of 1.70 ± 0.19 ppm (ozone dose floor value of 7.41 ppm · min). The efficacy of this treatment was higher at 76% RH compared to 34% RH, with notable differences after 50 min ($p < 0.01$) and 80 min ($p < 0.01$) of exposure time.

For an ozone exposure to 1.70 ± 0.19 ppm at 34% RH, the STMT (Fig 3D) did not increase airborne influenza A infectivity after 20 min ($p = 0.14$) and 50 min ($p = 0.29$) of exposure time. However, at 80 min of exposure ($p < 0.01$) (Fig 3D), the STMT helped protect the viral aerosols from the large (two orders of magnitude) decrease in infectivity that was observed for the NT (Fig 3C). At 76% RH, the viral aerosols exposed to the STMT exhibited a loss in infectivity of one order of magnitude after 20 min ($p = 0.01$) and 50 min ($p < 0.01$), meaning that the STMT did not protect aerosols from the ozone exposure of 1.70 ± 0.19 ppm. After 80 min, no difference in infectivity was observed between the NT and STMT aerosols ($p = 0.95$).

When combining the maximum infectivity loss observed from the reference condition (Fig 2; four orders of magnitude) and the ozone exposure (Fig 3; two orders of magnitude), the BPST resulted in the highest decrease of infectivity. Because the majority of the loss happened when influenza aerosols were exposed to the reference condition, the exposure to ozone becomes less powerful. Therefore, the combination of BSPT and ozone cannot be chosen as the more effective air treatment. The opposite was observed for the NT and STMT, with a maximum infectivity loss of one or four orders of magnitude when exposed to the reference condition and the ozone treatment, respectively. The high RH (76%) and longer exposure time (80 min) also exhibited the highest reduction of infectious influenza. Overall, an exposure of 1.70 ± 0.19 ppm of ozone at 76% RH for 80 min resulted in the highest infectivity loss for the NT and STMT.

## RSV infectivity assay

**Virus resistance in air sampler.**   The spiked RSV solution with 5% sucrose contained a concentration of $3.98 \times 10^5$ $TCID_{50}$/ml. The spiked RSV solution with 20% sucrose had a concentration of $1.26 \times 10^6$ $TCID_{50}$/ml.

After 20 minutes of sampling clean air with the SKC BioSampler, a loss of infectivity of one and two orders of magnitude was observed in the 5% and 20% sucrose solutions, respectively. The viral titers left in the two SKC BioSamplers with 5% sucrose were $3.98 \times 10^3$ $TCID_{50}$/ml and $2.24 \times 10^4$ $TCID_{50}$/ml, for a mean concentration of $1.32 \times 10^4$ $TCID_{50}$/ml. The viral titers that remained after sampling with a 20% sucrose solution were $1.26 \times 10^4$ $TCID_{50}$/ml and $7.08 \times 10^3$ $TCID_{50}$/ml, resulting in a mean concentration of $9.83 \times 10^3$ $TCID_{50}$/ml.

**Virus resistance in two nebulizers.**   The results from the infectivity assays performed with both the 6-jet Collison and the Aeroneb nebulizers are shown in Fig 4. The detection limit for the titration method was calculated with one positive well and corresponds to a concentration of $2.24 \times 10^1$ $TCID_{50}$/ml. The initial infectious concentration in the 6-jet Collison nebulizer was $5.87 \times 10^5 \pm 3.95 \times 10^5$ $TCID_{50}$/ml. After nebulization, the viral titer dropped considerably. Of the six replicates, two had concentrations of $3.98 \times 10^1$ $TCID_{50}$/ml, two others were at the detection limit of $2.24 \times 10^1$ $TCID_{50}$/ml, and the remaining two were below the detection limit. This suggests that when the RSV passes through the 6-jet Collison nebulizer, it seems to almost completely inactivate the virus.

When RSV infectivity was examined using the Aeroneb, the initial concentrations were $3.98 \times 10^5$ $TCID_{50}$/ml and $1.26 \times 10^6$ $TCID_{50}$/ml for the 5% and 20% sucrose solutions,

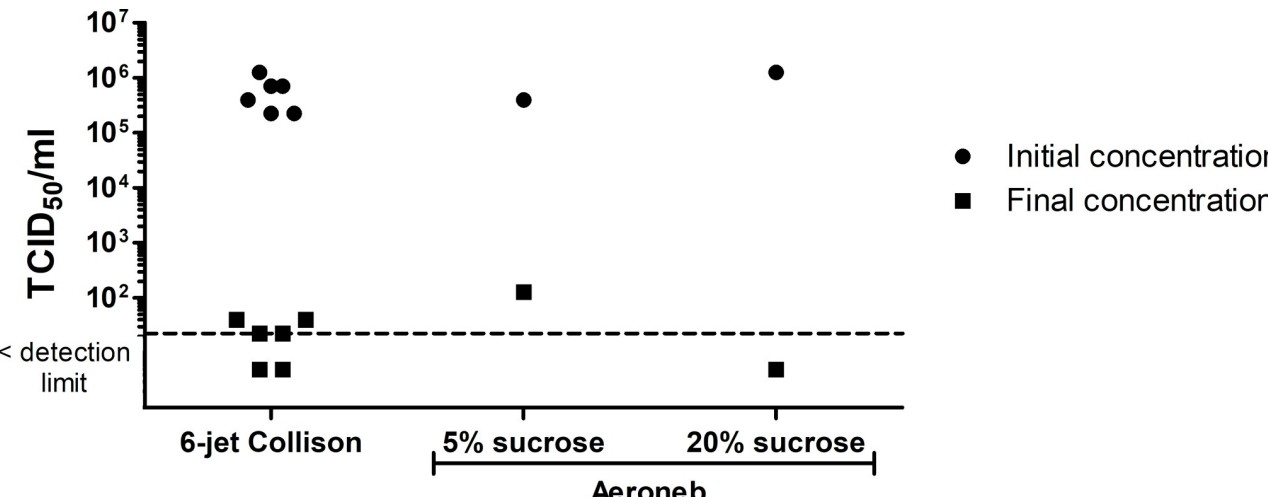

**Fig 4. Initial and final infectious RSV concentrations in the 6-jet Collison and the Aeroneb nebulizers.** The detection limit ($2.24 \times 10^1$ TCID$_{50}$/ml) is represented by the dotted line and corresponds to one positive well.

respectively. After nebulization, $1.26 \times 10^2$ TCID$_{50}$/ml remained infectious in the 5% sucrose solution, while the concentration was below the detection limit in the 20% sucrose solution. Therefore, despite the Aeroneb using a gentler nebulizing method, the RSV did not retain infectivity even with the sucrose supplement.

After assessing the results for RSV resistance to air sampling and nebulizing, it is clear that this virus does not maintain enough infectivity for reliable experimentation in the rotating aerosol chamber. For each condition, the combined infectivity loss from sampling and nebulizing was at least four orders of magnitude. Furthermore, aerosol loss is expected in the experimental setup, which can occur in the diffusion dryer tubes, at the junction of tubing and barb fittings and while sampling with the SKC BioSampler. From previous non-published experiments performed in our lab, it has been noted that there can be an aerosol loss of two to three orders of magnitude for every aerosol experiment. Consequently, the overall infectivity and aerosol loss would be around six orders of magnitude, which is too large to conduct experiments on the exposure to air and ozone for RSV, as was done for Influenza A.

## Discussion

The ozone probe used in this experiment allowed for the continued measurement of ozone concentrations inside the aerosol chamber at two RHs over a 70 min period (Fig 1) and the calculations of the ozone dose floor values. These additional informations obtained from the ozone probe are interesting since it was previously suggested that ozone interacts with water to form free radicals [38] and that high levels of humidity in the air could increase the production of free radicals and lead to virus inactivation [39, 40]. As seen in Fig 1 and Table 1, there is a notable difference in the initial ozone concentration measurements and dose floor values between the two tested RHs. Because all setup parameters were identical apart from the RH inside the chamber, one may conclude that the differences were caused by RH alone. The majority of the difference in ozone concentration between 34% and 76% also occurred during the first 10 min, which could indicate that the reaction with humidity is quite fast. Furthermore, the ozone concentration was still detectable at low RHs after 70 min, indicating that ozone was present during the entire exposure period. On the other hand, even if the ozone concentration was close to zero after 40 min at high RHs and that the ozone dose floor value

only increased from 6.89 ppm · min (40 min) to 7.41 ppm · min (70 min), the RIR values (Fig 3) show that the inactivation of airborne influenza was superior after longer exposure times. Additionally, for all tested conditions, the effect of ozone treatment is greater at high RHs than at low RHs, indicating that the free radicals that are produced have higher virucidal properties than ozone alone. This is consistent with previous studies [25, 40, 41]. Moreover, the addition of STM seems to have a protective effect at 34% RH after 80 min, while a disrupting effect is noted at 76% RH after 50 min. Although free radicals are produced when ozone interacts with water and organic compounds, this outcome could be the result of a synergy between high RH and the presence of proteins. Additional testing could be useful to verify if a synergistic inactivation effect is observed while using different supplements at high or low RH.

The NIRs that resulted from the NT and the STMT are comparable with those from Kormuth et al. (2018), in which HBE ECM was added. This confirmed that airborne influenza A maintains infectivity at low and high RHs for one hour [34]. However, our findings show that the BPST did not provide protection for influenza A aerosols. Indeed, it was responsible for NIR decreases of one to two (32% RH) and three to four (81% RH) orders of magnitude for the reference conditions (Fig 2B). One hypothesis for these results is that the surfactant interacts with the lipid membrane of the virions, which causes structural damage. This differs from the study by Vejerano and Marr (2018) study, which suggested that it could act as a protective viral coating [26]. When STM was added (Fig 2C), the infectivity loss was negligible when influenza A aerosols were exposed to the reference conditions. Influenza A therefore seems to be a resistant virus when airborne.

BPST experiments where influenza A aerosols were exposed to 0.23 ppm of ozone were carried out prior to the STMT experiments. NIR results indicated that the BPST led to more inactivated airborne influenza A compared to the NT. These observations led to the selection of the other supplement, STM, and exposure to a higher ozone concentration was not tested with the BPST.

We hypothesized that adding a supplement could cause a change in particle size, thus affecting the response of influenza A when aerosolized and altering its resistance to air or ozone. Particle MMADs were at $1.22 \pm 0.03$ μm at 32% RH and $1.26 \pm 0.03$ μm at 81% RH, nearly the same diameters as those observed when there was no supplement. The MMADs were also similar at low RHs for the BPST or STMT, at $1.18 \pm 0.03$ μm and $1.27 \pm 0.02$ μm, respectively. At high RHs, larger MMADs were observed for the BPST ($1.35 \pm 0.02$ μm) and the STMT ($1.51 \pm 0.07$ μm). The larger particle sizes do not explain the differences in infectivity that were observed as a result of exposure to the reference conditions (Fig 2), nor the differences observed due to ozone exposure (Fig 3).

RSV infectivity was assessed during two preliminary tests: the 6-jet Collison test and the infectivity assay. The information obtained from these tests was crucial because the experimental set up is not leak-proof and a loss of two to three orders of magnitude can be expected. Therefore, a minimum concentration of $10^5$–$10^6$ $TCID_{50}$/ml must be preserved in the nebulizing liquid in order to perform the exposure experiments for air and ozone. As shown in Fig 4, almost all RSV infectivity was lost during the nebulization step using the 6-jet Collison nebulizer. This nebulizer has been able to successfully aerosolize multiple viruses, including four phages [42], the murine norovirus (MNV-1) [24] and influenza A, suggesting that RSV may be more fragile than these other viruses. Since resistance behaviors from multiple aerosolized viruses are somewhat unpredictable [24, 43], the different behaviors of influenza A and RSV were not surprising. An infectivity assay verified whether the loss of infectivity was caused by the Aeroneb while nebulizing or the BioSampler while sampling. Despite the addition of sucrose to the RSV suspension, infectivity was lost through both devices. The number of infectious viruses that remained after being nebulized in the Aeroneb ($1.26 \times 10^2$ or $< 2.24 \times 10^1$

TCID$_{50}$/ml) was not sufficient to perform the subsequent exposure experiments. The SKC Bio-Sampler also resulted in a loss of infectivity of one to two orders of magnitude. Previous work from Grosz et al. (2014) showed that RSV maintains its infectivity when aerosolized with the PARI Sprint$^{TM}$ nebulizer [30]. This gentler device uses a compressor to produce aerosols instead of the vibrating membrane used in the Aeroneb, which could explain the higher infectious titers obtained in their study. In conclusion, RSV seems to be very sensitive to the nebulization processes of the 6-jet Collison and the Aeroneb nebulizers, as well as the sampling process associated with the SKC BioSampler.

Our experimental setup has been designed to allow us to work with level II viruses. Every part has been carefully chosen for that purpose, but compromises had to be made on some aspects. The aerosol chamber is constructed in aluminum to minimize particle loss due to static and it rotates to reduce particle deposition. It is also airtight and therefore maintains the level of relative humidity. Aerosol concentrations and MMAD over 18h have been studied in a previous publication [36]. The ozone is injected directly inside the chamber. The valves are shut after the injection to prevent the ozone from escaping the chamber. The ozone concentrations are then measured with a probe connected directly to the chamber. Consequently, the ozone concentration measures in the chamber are accurate. However, the aerosol path between the nebulizer and the chamber could lead to significant particle loss. As a safety measure, the material used to make the diffusion dryer tubes had to be shatterproof. It also had to be transparent to monitor the color of the silica beads. Therefore these tubes were made with polycarbonate. This plastic is known to have static properties, which could cause aerosol loss. These tubes are also not completely airtight because they protect the other parts of the setup by acting as breaking points in case of an accidental pressure increase. For the setup to fit inside the BSL II cabinet, the aerosol path includes 180˚ elbows in a few spots. The impaction of aerosols could occur in these elbows. The aerosol tubing used to connect the nebulizer, the diffusion dryers and the aerosol chamber is made of static dissipative silicon coated with carbon. However, since it the aerosol path (nebulizer–up to five diffusion dryers—chamber), particles could be lost by diffusion. As for the SKC BioSampler, this device preserves the virus infectivity and has a great collection efficiency of 1 μm particles but the re-aerosolization phenomenon has previously been documented [44]. While all these features could cause aerosol loss, we only evaluate the portion reaching the chamber by normalizing the infectious ratios. The drawbacks of this loss is that we are unable to test viruses that are sensitive or that do not produce high viral titers, as was the case for RSV.

The bacteriophage Phi6 has been used in multiple studies as a surrogate for influenza due to its similar RNA genome and envelope [24, 42, 45]. Phi6 was therefore a good candidate to be used as a model virus for RSV. The results obtained in our study for influenza A and RSV resistance to aerosolization and ozone exposure can be compared to those of Phi6. Dubuis et al. (2020) have shown that Phi6 loses its infectivity at low RH levels and is inactivated at high RHs when exposed to 1.13 ± 0.26 ppm of ozone [24]. In our study, even using 1.70 ± 0.19 ppm of ozone instead of 1.13 ± 0.26 ppm did not result in the inactivation of influenza A at low RHs. However, ozone was able to reduce its infectivity by four orders of magnitude at high RHs. Influenza A is significantly more resistant than Phi6 when aerosolized. Compared to RSV, which could not sustain nebulization and air sampling in our experiments, Phi6 is more robust when aerosolized. Therefore, the use of Phi6 as a surrogate for either airborne influenza A and RSV is not supported by our findings.

According to a guideline for virucidal chemical disinfectants published by the German Association for the Control of Virus Diseases and the Robert Koch Institute, a virucidal disinfectant must reduce viral infectivity by at least four orders of magnitude [46]. In this study, the effect of ozone as a disinfectant is presented using RIR values. For all the tested conditions, the

required infectivity reduction was achieved after exposure to 1.70 ± 0.19 ppm of ozone at 76% RH for 80 min (Fig 3C and 3D) which also corresponds to an ozone dose floor value of 7.41 ppm · min. The STMT did not protect airborne influenza A from the virucidal activity of ozone (Fig 3D), especially when compared to the NT (Fig 3C).

To the best of our knowledge, this study is the first to test an air disinfection protocol that uses ozone to inactivate airborne influenza A. The STMT has also provided great insight into its protective role for airborne influenza A. This supplement is only effective against ozone exposure at low RHs, and does not prevent the inactivation of aerosolized influenza A at high RHs.

The influenza virus is a persistent burden on hospitals, and air treatment could be an effective way to reduce its nosocomial transmission. The results from this study can be used as a basis for air decontamination in different settings. Since the studied ozone concentrations are above the threshold limit value–ceiling of 0.1 ppm [47], the decontamination protocol should be performed in leak-proof rooms in the absence of occupants for at least 80 min. This issue could be problematic for some settings that cannot evacuate the infected rooms for the proper amount of time required for air disinfection. Another element to consider is the reaction of ozone with some materials including rubber and its derivatives [48] which can cause the degradation of some mattresses [49]. Disinfection of recycled air inside the heating, ventilation and air-conditioning (HVAC) plenum could be an alternative to limit the material degradation and allow the occupants to stay in the rooms. One downside of this approach is the contact time, which cannot be performed over an 80-min period. The calculated ozone dose floor value required to achieve a reduction of four orders of magnitude was 7.41 ppm·min and was obtained at 76% RH. Since the real ozone dose to which viruses were exposed might be significantly higher than the calculated ozone dose floor value, the exposure in the HVAC plenum should aim for an ozone dose of at least 7.41 ppm·min at an RH close to 76%. Depending on the contact time in the HVAC plenum, ozone generation could be modulated to achieve this minimal dose. The other limitation of a disinfection in the HVAC plenum is that the furniture and areas hard to reach inside the infected rooms would not be decontaminated. Considering these elements, each setting should assess which option is the most appropriate to meet its needs before implementing an air disinfection protocol using ozone.

The emergence of the SARS-CoV-2 pandemic has shed light on the transmission of viruses through air. Evidence suggests that aerosols in hospital rooms can contain SARS-CoV-2 viruses [50, 51], making the implementation of ozone air treatments a practical way to reduce viral transmission and to ensure a safe environment for patients. Although ozone disinfection has not been tested directly on SARS-CoV-2 aerosols, evidence from the inactivation of other airborne viruses (four phages, murine norovirus and influenza) suggest that ozone could be a useful disinfectant for different types of viruses. As further evidence, a recent study has suggested that ozone could be a potential disinfectant for SARS-CoV-2 by targeting its spike protein and envelope lipids [52].

## Conclusion

In conclusion, the efficacy of an air treatment protocol using ozone to target airborne influenza A was evaluated. RSV did not survive the aerosolization and sampling processes in our laboratory setup, and therefore ozone exposure experiments could not be conducted using this virus. In order to better reflect aerosols produced by infected people, which can contain surfactants and proteins, the nebulizing liquid was supplemented with BPS and STM for experimentation. These various conditions provided clues to understanding the protective roles of BPS and STM for influenza A when aerosolized and exposed to ozone. The air treatment that resulted

in a reduction in infectivity of four orders of magnitude was the exposure to 1.70 ± 0.19 ppm of ozone at 76% RH for 80 min, and was effective for both the STMT and the NT, indicating that ozone could be qualified as a virucidal disinfectant.

This study provides robust results regarding the efficacy of air treatment, which could be useful for the control of nosocomial transmission of influenza A. Further tests should be conducted in order to determine how air disinfection could be implemented in hospital settings and other environments that could benefit from this technology.

## Supporting information

**S1 File. Regression analyses.**
(DOCX)

## Acknowledgments

The authors would like to thank Liva Checkmahomed and Marie-Eve Hamelin for technical support as well as Serge Simard for statistical analysis and Amanda Toperoff and Michi Waygood for English revision.

## Author Contributions

**Conceptualization:** Marie-Eve Dubuis, Nathalie Turgeon, Nathalie Grandvaux, Caroline Duchaine.

**Data curation:** Marie-Eve Dubuis.

**Formal analysis:** Marie-Eve Dubuis, Étienne Racine.

**Funding acquisition:** Nathalie Grandvaux, Caroline Duchaine.

**Investigation:** Marie-Eve Dubuis, Jonathan M. Vyskocil, Christophe Tremblay, Espérance Mukawera.

**Methodology:** Marie-Eve Dubuis, Nathalie Grandvaux, Caroline Duchaine.

**Project administration:** Marie-Eve Dubuis, Nathalie Grandvaux, Caroline Duchaine.

**Resources:** Guy Boivin, Nathalie Grandvaux, Caroline Duchaine.

**Supervision:** Nathalie Turgeon, Guy Boivin, Nathalie Grandvaux, Caroline Duchaine.

**Validation:** Caroline Duchaine.

**Visualization:** Marie-Eve Dubuis.

**Writing – original draft:** Marie-Eve Dubuis.

**Writing – review & editing:** Étienne Racine, Jonathan M. Vyskocil, Nathalie Turgeon, Christophe Tremblay, Guy Boivin, Nathalie Grandvaux, Caroline Duchaine.

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
