## [Decision Letter · Decision Letter 0]

8 Apr 2021

PONE-D-21-02103

Ozone inactivation of airborne influenza and lack of resistance of respiratory syncytial virus to aerosolization and sampling processes

PLOS ONE

Dear Dr. Duchaine,

Thank you for submitting your manuscript to PLOS ONE. After careful consideration, we feel that it has merit but does not fully meet PLOS ONE’s publication criteria as it currently stands. Therefore, we invite you to submit a revised version of the manuscript that addresses the points raised during the review process.

We look forward to receiving your revised manuscript.

Kind regards,

Sander Herfst

Academic Editor

PLOS ONE

2. Please provide the source of the Influenza A/Michigan/45/2015 sample you used in this study.

Reviewers' comments:

Reviewer's Responses to Questions

**Comments to the Author**

1. Is the manuscript technically sound, and do the data support the conclusions?

Reviewer #1: Yes

Reviewer #2: Partly

Reviewer #3: Yes

2. Has the statistical analysis been performed appropriately and rigorously? 

Reviewer #1: Yes

Reviewer #2: Yes

Reviewer #3: Yes

3. Have the authors made all data underlying the findings in their manuscript fully available?

Reviewer #1: No

Reviewer #2: Yes

Reviewer #3: Yes

4. Is the manuscript presented in an intelligible fashion and written in standard English?

Reviewer #1: Yes

Reviewer #2: Yes

Reviewer #3: Yes

5. Review Comments to the Author

Reviewer #1: The authors of this manuscript elaborated on previous work that described the inactivation of aerosolized phages and murine norovirus by ozone (Dubuis et al., 2020). Here the authors used the same approach to study the inactivation of an influenza- and respiratory syncytial virus. This study provides new insights in the aerosol stability of influenza viruses in relation with relative humidity and ozone concentration, which are placed in perspective with other studies. However, some points are unclear to the reviewer and need to be clarified prior to publication.

1. What is the composition of the non-treated nebulizing liquid (is it a buffer or a medium?) and do the authors expect hardly any inactivation after incubation in air for 60 min under these conditions?

2. The nebulizing liquid is supplemented with either BPS or STM, of which the authors stated that this was as previously done by Kormuth et al., 2018 with extra cellular material from human bronchial epithelial cells (HBE ECM) (Line 108). However although 10% supplementation is used in both studies, this does not necessarily result in comparable conditions in terms of e.g. protein/lipid content. STM for example contain albumin and pig gastric mucin, which are absent in the HBE ECM. Can the authors elaborate more on the rationale behind the supplementation of the nebulizing liquid?

3. The inactivation of the viruses are all presented as normalized infectious ratio’s or relative infectious ratio’s, which is understandable as two variables are tested: treatment of the nebulizing liquid and the effect of ozone on virus inactivation under each condition. The downside of this approach is however that the reader does not have any idea on the number of virus particles that were nebulized and were collected in each experiment and whether these numbers were comparable between the different experiments. The reviewer would recommend to show this data at least for the reference conditions for more clarity.

4. Line 279: ‘addition of BPS did not offer additional protection of airborne influenza A …”. This sentence is quite odd in the reviewers opinion as the authors previously showed that BPS affects virus infectivity with 2-3 logs (Fig. 2B) upon aging in air in absence of ozone.

5. Can the authors explain why the addition of STM is protective after 80 min expose at an RH of 34%, but resulted more inactivation at a RH of 75% after 50 minute exposure?

Textual comments

Line 72: Maybe specify the droplet size/volume for more clarity

Line 285-286, please mention which ozone dose for clarity.

Line 289: Maybe it is good to make immediately clear that this observation is obtained at 1.7 ppm ozone. Now it is stated at the end.

Line 406-407. Can the authors provide references for this claim?

Reviewer #2: The Dubuis et al. manuscript describes efforts to evaluate the virucidal efficacy of ozone against Influenza A and Respiratory Syncytial Virus (RSV) containing aerosols. This addresses the important topic of methods to reduce nosocomial transmission of respiratory viruses. The authors utilize a test system encompassing a 6-jet collison nebulizer introducing virus containing aerosols into a rotating drum aerosol chamber with periodic sampling with an SKC BioSampler. The authors show that 1.7+/-0.19ppm ozone at 76% relative humidity (RH) for 80 minutes resulted in a four orders of magnitude loss of infectious Influenza A. They conclude that this treatment could be used as a virucidal disinfectant for the control of nosocomial transmission of Influenza A. However, additional efforts to ensure that the loss in infectivity of Influenza A is due to ozone inactivation and not through losses to the system are required. Also, the addition of RSV aerosolization and sampling efforts adds confusion to the manuscript and disrupts its flow.

Major Issues:

(1) The authors were able to achieve sufficient detection of infectious Influenza A using the collison nebulizer and SKC sampler system to evaluate ozone treatment as an aerosol disinfection method. However, the authors were unable to achieve such success in detecting sufficient amounts of infectious RSV to perform similar evaluations. While the authors made progress on evaluating the stability of RSV during aerosolization and sampling, much work needs to be done in further characterizing aerosolization and sampling methods for publication. The addition of these efforts adds unnecessary confusion for the readers and detracts from the Influenza A findings.

(2) In the discussion, the authors state that “the experimental set up is not leak-proof and a loss of two to three orders of magnitude can be expected.” The authors should consider additional methods to reduce and/or control this loss for several reasons. The loss in recovery over time should be reported directly, at least as a representative plot, to show readers that a significant amount of virus is lost over time due to physical loss to the chamber wall and leaks, and the loss of infectivity of Influenza A in aerosols (which has been reported in Schuit et al. JID 2020). Adding this data would also help in detailing the range in concentration of infectious virus that is being used to calculate the NIR and RIR ratios. The leakiness of the system is also likely contributing to the significant decrease in ozone levels over time in the chamber. While it is likely unavoidable in a hospital setting, ensuring the correct concentration of ozone in a closed, experimental setup is crucial before attempting to implement this type of strategy in a real-world scenario. While the authors take efforts to normalize these losses to the experimental setup through PCR and concentration within the nebulizer at the start of the experiments, additional efforts must be made to ensure that the loss in infectivity is due to ozone.

Minor Issues:

(1) The concentration of ozone at 76% RH is noted to approach zero after 30 minutes, however, the experiments with the greatest efficiency in inactivation are carried out for 80 minutes. Also, the mean RIR for NT +1.70+/-0.19ppm at 76% RH for 80 minutes is much lower than that of 50 minutes, even though the ozone concentrations at these time points is negligible, if present at all. Additional discussion, if meaningful, should be included to rationalize the impact of ozone at these longer time-points when the concentration are so low, especially in light of point 2 above.

Reviewer #3: Comments/Critiques

- Is there a reason that ozone concentration and ozone exposure time are not combined into a single "dose" parameter as is used in Chick-Watson-type considerations of disinfection? This is the approach used by Tseng and Li, which is cited in the manuscript. Such a combined parameter seems particularly important in this study since the ozone concentrations were not constant and decreased over time. Therefore trials listed in the manuscript, for example, as representing aerosol exposure to 1.7 ppm of ozone are actually exposures starting at 1.7 ppm and gradually and continuously decreasing over 70 min to less than 0.3 ppm. It is the dose - the integral under this concentration-time curve - that should be reported and should be used to infer the resulting effects.

- Line 350: In discussing chamber testing, is it an accurate description to say "the chamber leaks"? Is there evidence of this that can be referenced? Is the chamber leaking or are the aerosols simply being lost to the chamber interior surfaces? The assertion of a leak in a process intended to conserve aerosol mass is troubling but can be easily assessed using a tracer gas. Such assessment is recommended if the authors maintain the view that chamber leakage is a systematic bias of the system.

- Although the study cites the opportunity to introduce air treatments for preventing the spread of airborne infectious viral diseases as its motivation, the manuscript does not suggest what form such treatments might take. The ozone concentrations examined in the study are higher than generally allowed for human occupied spaces, and the exposure times examined in the study are generally longer than would be required to evacuate a(n unoccupied) room of contaminated air using the ventilation system (hospital rooms operate at ventilation rates comparable to several complete changes of room air per hour). If the motivation is air treatment, then the manuscript should provide some discussion of how this might be implemented in a way that is at least comparable to these incumbent approaches. Beyond this minimum, there are material degradation concerns that can limit ozone use that may not be most suitable for this manuscript, but are of concern and warrant some mention.

Minor Critiques/Typographic and Grammatical Errors:

- Line 59: The previous sentence that focuses on influenza makes clear that the authors' inference about airborne transmission stems from data resulting from measurement of airborne samples. However, for RSV (starting at Line 59), none of the various sample locations described refer to airborne samples; therefore, in the next sentence ("Only a small number....") it is not clear whether the authors are arguing that airborne transmission is unlikely due to the scarcity of *airborne* samples or whether airborne transmission is unlikely due to the scarcity of *any* samples. Please clarify.

- Line 63: Manuscript would be improved if additional context were given to the characterization of "high" used to describe genome copies per cubic meter of air. Clearly, this depends on distance from the source, air changes per hour imposed by ventilation in the room, etc.

- Line 71: Similar critique as above: the characterization of "low concentrations of ozone" would be greatly improved if additional context were provided. There are multiple contexts to consider. There is the context of prior studies of ozonation for sterilization as well as the limits on ozone concentrations in air for protection of human health. The latter is much lower and therefore likely would undercut any unqualified assertion that the ozone concentrations of the present studies are "low".

- Line 120: The sentence beginning "Diffusion dryer tubes...." is missing a verb.

- Line 133: Dilute (i.e., ppm-level) concentrations of ozone in air are inaccurately described as "ozone" (solely).

- Line 145: Why was a new ozone monitor introduced?

- Line 148: The second phrase in this sentence lacks a verb ("....which 20 ppb and 50 ppb, with .....")

6. PLOS authors have the option to publish the peer review history of their article (what does this mean?). If published, this will include your full peer review and any attached files.

Reviewer #1: No

Reviewer #2: No

Reviewer #3: No

---

## [Author Response · Author response to Decision Letter 0]

21 May 2021

Dear Editor and Reviewers

We are very thankful for the thorough revision of our manuscript and fruitful comments provided. Here is a point-by-point response to each reviewers’ comments and details about the changes made. We hope this revised version meets your expectations.

RESPONSE: We have updated the style requirements according to the templates. We also have renamed the figure files.

2. Please provide the source of the Influenza A/Michigan/45/2015 sample you used in this study.

RESPONSE: We have added the reference “Influenza A/Michigan/45/2015 (clinical strain A/Quebec/22578/2016 provided by Guy Boivin) was propagated in MDCK (CCL-34, ATCC) and ST6GalI-MDCK cells.”

Reviewers' comments:

Reviewer's Responses to Questions

Comments to the Author

Reviewer #1: The authors of this manuscript elaborated on previous work that described the inactivation of aerosolized phages and murine norovirus by ozone (Dubuis et al., 2020). Here the authors used the same approach to study the inactivation of an influenza- and respiratory syncytial virus. This study provides new insights in the aerosol stability of influenza viruses in relation with relative humidity and ozone concentration, which are placed in perspective with other studies. However, some points are unclear to the reviewer and need to be clarified prior to publication.

1. What is the composition of the non-treated nebulizing liquid (is it a buffer or a medium?) and do the authors expect hardly any inactivation after incubation in air for 60 min under these conditions?

RESPONSE: The nebulizing liquid is a viral lysate from the cell culture. We have added this precision in the sections “influenza A virus and host cells” and “Influenza A aerosolization, exposure to ozone and sampling”. 

2. The nebulizing liquid is supplemented with either BPS or STM, of which the authors stated that this was as previously done by Kormuth et al., 2018 with extra cellular material from human bronchial epithelial cells (HBE ECM) (Line 108). However although 10% supplementation is used in both studies, this does not necessarily result in comparable conditions in terms of e.g. protein/lipid content. STM for example contain albumin and pig gastric mucin, which are absent in the HBE ECM. Can the authors elaborate more on the rationale behind the supplementation of the nebulizing liquid?

RESPONSE: Since the aerosols produced by the human respiratory tract contain other elements such as protein, lipids and salts, we wanted to represent this content when aerosolizing viruses. We initially thought that the droplet’s contents concentrated into the droplet nuclei could provide a protection from ozone. Therefore, aerosolizing viruses from lysate would not be as representative as real aerosols produced by infected humans. We agree with your comment that the STM main ingredients are not the same as the HBE ECM composition. However, we needed almost 60 ml of this supplement to conduct our experiments, which is a volume way above what we could produce by pooling 150 µl washes of HBE ECM as performed by Kormuth et al. (2018). Therefore, we had to find a solution that could be made in higher volumes. The STM created by Hamed and Fiegel (2014) has chemical and physical properties similar to native mucus and we could prepare 60 ml for our experiments. We added a sentence in the “Influenza A aerosolization, exposure to ozone and sampling” section: “However, we could not use pooled washes of HBE ECM because a volume of 60 ml of the supplement was needed for our experiments.”

3. The inactivation of the viruses are all presented as normalized infectious ratio’s or relative infectious ratio’s, which is understandable as two variables are tested: treatment of the nebulizing liquid and the effect of ozone on virus inactivation under each condition. The downside of this approach is however that the reader does not have any idea on the number of virus particles that were nebulized and were collected in each experiment and whether these numbers were comparable between the different experiments. The reviewer would recommend to show this data at least for the reference conditions for more clarity.

RESPONSE: Thank you for your comment. We added this specification in the reference conditions section. “It is worth mentioning that the nebulized concentrations of infectious viruses for the NT, BPST and STMT experiments were 6.96 X 107 ± 9.40 X 107 PFU/ml, 1.58 X 107 ± 5.35 X 106 PFU/ml and 1.58 X 107 ± 4.54 X 106 PFU/ml, respectively. The concentrations of sampled viruses were lower, at 1.58 X 105 ± 1.92 X 105 PFU/ml for the NT, 2.20 X 104 ± 3.49 X 104 for the BPST and 1.58 X 104 ± 2.26 X 104 for the STMT.” 

4. Line 279: ‘addition of BPS did not offer additional protection of airborne influenza A …”. This sentence is quite odd in the reviewers opinion as the authors previously showed that BPS affects virus infectivity with 2-3 logs (Fig. 2B) upon aging in air in absence of ozone.

RESPONSE: We understand your concern. Here is the new sentence: “This means that the BPS did not protect the remaining infectious airborne influenza A from an exposure to 0.23 ppm ± 0.03 ppm of ozone.”

5. Can the authors explain why the addition of STM is protective after 80 min expose at an RH of 34%, but resulted more inactivation at a RH of 75% after 50 minute exposure?

RESPONSE: We agree that this observation deserves an explanation. We have added these sentences in the discussion at the end of the first paragraph: “Moreover, the addition of STM seems to have a protective effect at 34% RH after 80 min, while a disrupting effect is noted at 75% RH after 50 min. Although free radicals are produced when ozone interacts with water and organic compounds, this outcome could be the result of a synergy between high RH and the presence of proteins. Additional testing could be useful to verify if a synergistic inactivation effect is observed while using different supplements at high or low RH.”

Textual comments

6. Line 72: Maybe specify the droplet size/volume for more clarity

RESPONSE: We have added these elements to the article. “According to Vejerano and Marr (2018), a single human-produced evaporating respiratory droplet of 60 µm (roughly 0.1 nl) is estimated to compose 1 ng of salt, 1 ng of protein and 0.06 ng of surfactant.” 

7. Line 285-286, please mention which ozone dose for clarity.

RESPONSE: We have added this information. “The maximum reduction in infectivity was four orders of magnitude, which was obtained after 80 min at 76% RH for an ozone exposure of 1.70 ± 0.19 ppm.” 

8. Line 289: Maybe it is good to make immediately clear that this observation is obtained at 1.7 ppm ozone. Now it is stated at the end.

RESPONSE: We have added the concentration at the beginning of the paragraph. “For an ozone exposure to 1.70 ± 0.19 ppm at 34% RH, the STMT (Fig 3D) did not increase airborne influenza A infectivity after 20 min (p = 0.14) and 50 min (p = 0.29) of exposure time.”

9. Line 406-407. Can the authors provide references for this claim?

RESPONSE: We have modified this sentence and included references for the new claim: “Since resistance behaviors from multiple aerosolized viruses are somewhat unpredictable [24, 43], the different behaviors of influenza A and RSV were not surprising.”

Reviewer #2: The Dubuis et al. manuscript describes efforts to evaluate the virucidal efficacy of ozone against Influenza A and Respiratory Syncytial Virus (RSV) containing aerosols. This addresses the important topic of methods to reduce nosocomial transmission of respiratory viruses. The authors utilize a test system encompassing a 6-jet collison nebulizer introducing virus containing aerosols into a rotating drum aerosol chamber with periodic sampling with an SKC BioSampler. The authors show that 1.7+/-0.19ppm ozone at 76% relative humidity (RH) for 80 minutes resulted in a four orders of magnitude loss of infectious Influenza A. They conclude that this treatment could be used as a virucidal disinfectant for the control of nosocomial transmission of Influenza A. However, additional efforts to ensure that the loss in infectivity of Influenza A is due to ozone inactivation and not through losses to the system are required. Also, the addition of RSV aerosolization and sampling efforts adds confusion to the manuscript and disrupts its flow.

Major Issues:

1. The authors were able to achieve sufficient detection of infectious Influenza A using the collison nebulizer and SKC sampler system to evaluate ozone treatment as an aerosol disinfection method. However, the authors were unable to achieve such success in detecting sufficient amounts of infectious RSV to perform similar evaluations. While the authors made progress on evaluating the stability of RSV during aerosolization and sampling, much work needs to be done in further characterizing aerosolization and sampling methods for publication. The addition of these efforts adds unnecessary confusion for the readers and detracts from the Influenza A findings.

RESPONSE: Thank you for your insight. We understand that the results obtained for RSV are not as satisfying as those with Influenza. However, we strongly feel that it is important to present negative or inconclusive results since they have scientific value. They also provide a comparison between influenza and RSV. This kind of comparison is not described in the literature and this information could be really useful for other teams who would like to study airborne RSV.

2. In the discussion, the authors state that “the experimental set up is not leak-proof and a loss of two to three orders of magnitude can be expected.” The authors should consider additional methods to reduce and/or control this loss for several reasons. The loss in recovery over time should be reported directly, at least as a representative plot, to show readers that a significant amount of virus is lost over time due to physical loss to the chamber wall and leaks, and the loss of infectivity of Influenza A in aerosols (which has been reported in Schuit et al. JID 2020). Adding this data would also help in detailing the range in concentration of infectious virus that is being used to calculate the NIR and RIR ratios. The leakiness of the system is also likely contributing to the significant decrease in ozone levels over time in the chamber. While it is likely unavoidable in a hospital setting, ensuring the correct concentration of ozone in a closed, experimental setup is crucial before attempting to implement this type of strategy in a real-world scenario. While the authors take efforts to normalize these losses to the experimental setup through PCR and concentration within the nebulizer at the start of the experiments, additional efforts must be made to ensure that the loss in infectivity is due to ozone.

RESPONSE: Since another reviewer also made a comment regarding the leakages, we have decided addressed both requests at the same time. We have modified the paragraph describing the issue of aerosol losses in the experimental set up (results section) but we have not given too much details there. We have added a paragraph in the discussion explaining the pros and cons of our experimental set up so that the reader can understand why some materials were chosen and why we cannot do some experiences using sensitive viruses. We have also addressed the issue of the ozone concentration.

“Furthermore, aerosol loss is expected in the experimental setup, which can occur in the diffusion dryer tubes, at the junction of tubing and barb fittings and while sampling with the SKC BioSampler. From previous non-published experiments performed in our lab, it has been noted that there can be an aerosol loss of two to three orders of magnitude for every aerosol experiment.”

“Our experimental setup has been designed to allow us to work with level II viruses. Every part has been carefully chosen for that purpose, but compromises had to be made on some aspects. The aerosol chamber is constructed in aluminum to minimize particle loss due to static and it rotates to reduce particle deposition. It is also airtight and therefore maintains the level of relative humidity. Aerosol concentrations and MMAD over 18h have been studied in a previous publication [36]. The ozone is injected directly inside the chamber. The valves are shut after the injection to prevent the ozone from escaping the chamber. The ozone concentrations are then measured with a probe connected directly to the chamber. Consequently, the ozone concentration measures in the chamber are accurate. However, the aerosol path between the nebulizer and the chamber could lead to significant particle loss. As a safety measure, the material used to make the diffusion dryer tubes had to be shatterproof. It also had to be transparent to monitor the color of the silica beads. Therefore these tubes were made with polycarbonate. This plastic is known to have static properties, which could cause aerosol loss. These tubes are also not completely airtight because they protect the other parts of the setup by acting as breaking points in case of an accidental pressure increase. For the setup to fit inside the BSL II cabinet, the aerosol path includes 180° elbows in a few spots. The impaction of aerosols could occur in these elbows. The aerosol tubing used to connect the nebulizer, the diffusion dryers and the aerosol chamber is made of static dissipative silicon coated with carbon. However, since it the aerosol path (nebulizer – up to five diffusion dryers - chamber), particles could be lost by diffusion. As for the SKC BioSampler, this device preserves the virus infectivity and has a great collection efficiency of 1 µm particles but the re-aerosolization phenomenon has previously been documented [43]. While all these features could cause aerosol loss, we only evaluate the portion reaching the chamber by normalizing the infectious ratios. The drawbacks of this loss is that we are unable to test viruses that are sensitive or that do not produce high viral titers, as was the case for RSV.”

Minor Issues:

3. The concentration of ozone at 76% RH is noted to approach zero after 30 minutes, however, the experiments with the greatest efficiency in inactivation are carried out for 80 minutes. Also, the mean RIR for NT +1.70+/-0.19ppm at 76% RH for 80 minutes is much lower than that of 50 minutes, even though the ozone concentrations at these time points is negligible, if present at all. Additional discussion, if meaningful, should be included to rationalize the impact of ozone at these longer time-points when the concentration are so low, especially in light of point 2 above.

RESPONSE: We have modified one sentence in the discussion to point out this outcome. “On the other hand, even if the ozone concentration was close to zero after 40 min at high RHs, the RIR values (Fig 3) show that the inactivation of airborne influenza was superior after longer exposure times. Additionally, for all tested conditions, the effect of ozone treatment is greater at high RHs than at low RHs, indicating that the free radicals that are produced have higher virucidal properties than ozone alone.”

Reviewer #3: Comments/Critiques

1. Is there a reason that ozone concentration and ozone exposure time are not combined into a single "dose" parameter as is used in Chick-Watson-type considerations of disinfection? This is the approach used by Tseng and Li, which is cited in the manuscript. Such a combined parameter seems particularly important in this study since the ozone concentrations were not constant and decreased over time. Therefore trials listed in the manuscript, for example, as representing aerosol exposure to 1.7 ppm of ozone are actually exposures starting at 1.7 ppm and gradually and continuously decreasing over 70 min to less than 0.3 ppm. It is the dose - the integral under this concentration-time curve - that should be reported and should be used to infer the resulting effects.

RESPONSE: Thank you for pointing this out. We decided to use concentration values instead of single doses for field application purposes. The initial (and highest) concentration of 1.70 ppm is important for healthcare settings because of the impacts of ozone exposure on human health. Furthermore, ozone exposure for humans is always reported in concentration and not with a dose. 

As you suggested, we have calculated the ozone doses using the integral under the concentration-time curve. These new results are below the concentration results (Fig 1) in the “Ozone concentrations and doses” section. We present the doses in a table (Table 1) with the appropriate explanations. Because these are new results, we have added a paragraph in the “Calculations and statistical analysis” section and have also added sentences in the discussion. We have uploaded a supporting information file with the regression analyses information. Here are the main additions to the manuscript:

A. Calculations and statistical analysis: Ozone doses were estimated by calculating the area under concentration-vs-time curves, for RH values of 34% and 76%. Concentration-vs-time curves were fitted to observed data by regression analysis for exposure times between 10 min and 70 min (see supporting information for details). Regression analyses were performed in RStudio (version 1.2.5033). The areas under the curves were calculated numerically with a trapezoid scheme coded in C++ and validated against analytical computations. For exposure times between 0 and 10 min, extrapolation of the fitted concentration-vs-time curves was deemed too unreliable for dose calculations. Accumulated dose between 0 and 10 min was instead estimated by assuming that ozone concentration was constant over that time interval and equal to the fitted concentration value at 10 min. Consequently, calculated ozone doses represent floor values. Confidence intervals were not reported for dose calculations because the uncertainty associated to the constant concentration assumption was much larger than the width of the confidence bands around the regressions curves.

B. Ozone concentrations and doses: The calculated ozone doses floor values for both RH levels at 10-minute intervals are presented in Table 1. Even if the ozone injection was identical, after an exposure time of 70 min, the ozone dose floor value at 34% was much higher, at 58.67 ppm ∙ min, than the dose at 76%, at 7.41 ppm ∙ min. Furthermore, the values increase by a factor of two between 10 and 70 minutes at 76%, while they increase by a factor of three at 34% RH. 

Table 1. Ozone doses floor value (ppm ∙ min) for each exposure time at 34% and 76% RH. 

Exposure Time (min) Ozone dose (ppm ∙ min) 

 34% RH 76% RH

10 16.90 3.69

20 31.11 5.72

30 41.05 6.49

40 48.01 6.89

50 52.88 7.13

60 56.29 7.29

70 58.67 7.41

C. Discussion: The calculated ozone dose floor value required to achieve a reduction of four orders of magnitude was 7.41 ppm∙min and was obtained at 76% RH. Since the real ozone dose to which viruses were exposed might be significantly higher than the calculated ozone dose floor value, the exposure in the HVAC plenum should aim for an ozone dose of at least 7.41 ppm∙min at an RH close to 76%. Depending on the contact time in the HVAC plenum, ozone generation could be modulated to achieve this minimal dose.

2. Line 350: In discussing chamber testing, is it an accurate description to say "the chamber leaks"? Is there evidence of this that can be referenced? Is the chamber leaking or are the aerosols simply being lost to the chamber interior surfaces? The assertion of a leak in a process intended to conserve aerosol mass is troubling but can be easily assessed using a tracer gas. Such assessment is recommended if the authors maintain the view that chamber leakage is a systematic bias of the system.

RESPONSE: We agree with your comment that there are differences between leaks and loss. Another reviewer made a comment regarding the leaks, therefore we have addressed both requests at the same time. We have modified the paragraph describing the issue of aerosol losses in the experimental set up (results section) but we have not given too much details there. We have added a paragraph in the discussion explaining the pros and cons of our experimental set up so that the reader can understand why some materials were chosen and why we cannot do some experiences using sensitive viruses. We have also addressed the issue of the ozone concentration.

“Furthermore, aerosol loss is expected in the experimental setup, which can occur in the diffusion dryer tubes, at the junction of tubing and barb fittings and while sampling with the SKC BioSampler. From previous non-published experiments performed in our lab, it has been noted that there can be an aerosol loss of two to three orders of magnitude for every aerosol experiment.”

“Our experimental setup has been designed to allow us to work with level II viruses. Every part has been carefully chosen for that purpose, but compromises had to be made on some aspects. The aerosol chamber is constructed in aluminum to minimize particle loss due to static and it rotates to reduce particle deposition. It is also airtight and therefore maintains the level of relative humidity. Aerosol concentrations and MMAD over 18h have been studied in a previous publication [36]. The ozone is injected directly inside the chamber. The valves are shut after the injection to prevent the ozone from escaping the chamber. The ozone concentrations are then measured with a probe connected directly to the chamber. Consequently, the ozone concentration measures in the chamber are accurate. However, the aerosol path between the nebulizer and the chamber could lead to significant particle loss. As a safety measure, the material used to make the diffusion dryer tubes had to be shatterproof. It also had to be transparent to monitor the color of the silica beads. Therefore these tubes were made with polycarbonate. This plastic is known to have static properties, which could cause aerosol loss. These tubes are also not completely airtight because they protect the other parts of the setup by acting as breaking points in case of an accidental pressure increase. For the setup to fit inside the BSL II cabinet, the aerosol path includes 180° elbows in a few spots. The impaction of aerosols could occur in these elbows. The aerosol tubing used to connect the nebulizer, the diffusion dryers and the aerosol chamber is made of static dissipative silicon coated with carbon. However, since it the aerosol path (nebulizer – up to five diffusion dryers - chamber), particles could be lost by diffusion. As for the SKC BioSampler, this device preserves the virus infectivity and has a great collection efficiency of 1 µm particles but the re-aerosolization phenomenon has previously been documented [43]. While all these features could cause aerosol loss, we only evaluate the portion reaching the chamber by normalizing the infectious ratios. The drawbacks of this loss is that we are unable to test viruses that are sensitive or that do not produce high viral titers, as was the case for RSV.”

3. Although the study cites the opportunity to introduce air treatments for preventing the spread of airborne infectious viral diseases as its motivation, the manuscript does not suggest what form such treatments might take. The ozone concentrations examined in the study are higher than generally allowed for human occupied spaces, and the exposure times examined in the study are generally longer than would be required to evacuate a(n unoccupied) room of contaminated air using the ventilation system (hospital rooms operate at ventilation rates comparable to several complete changes of room air per hour). If the motivation is air treatment, then the manuscript should provide some discussion of how this might be implemented in a way that is at least comparable to these incumbent approaches. Beyond this minimum, there are material degradation concerns that can limit ozone use that may not be most suitable for this manuscript, but are of concern and warrant some mention.

RESPONSE: Thank you for your comment. We have added a paragraph in the discussion debating the kind of air treatment protocols that could be implemented (pros and cons). We have also added the calculated ozone dose floor value that would be required to achieve a reduction of four orders of magnitude. Here is the new paragraph: “The Influenza virus is a persistent burden on hospitals, and air treatment could be an effective way to reduce its nosocomial transmission. The results from this study can be used as a basis for air decontamination in different settings. Since the studied ozone concentrations are above the threshold limit value – ceiling of 0.1 ppm [47], the decontamination protocol should be performed in leak-proof rooms in the absence of occupants for at least 80 min. This issue could be problematic for some settings that cannot evacuate the infected rooms for the proper amount of time required for air disinfection. Another element to consider is the reaction of ozone with some materials including rubber and its derivatives [48] which can cause the degradation of some mattresses [49]. Disinfection of recycled air inside the heating, ventilation and air-conditioning (HVAC) plenum could be an alternative to limit the material degradation and allow the occupants to stay in the rooms. One downside of this approach is the contact time, which cannot be performed over an 80-min period. The calculated ozone dose floor value required to achieve a reduction of four orders of magnitude was 7.41 ppm∙min and was obtained at 76% RH. Since the real ozone dose to which viruses were exposed might be significantly higher than the calculated ozone dose floor value, the exposure in the HVAC plenum should aim for an ozone dose of at least 7.41 ppm∙min at an RH close to 76%. Depending on the contact time in the HVAC plenum, ozone generation could be modulated to achieve this minimal dose. The other limitation of a disinfection in the HVAC plenum is that the furniture and areas hard to reach inside the infected rooms would not be decontaminated. Considering these elements, each setting should assess which option is the most appropriate to meet its needs before implementing an air disinfection protocol using ozone.”

Minor Critiques/Typographic and Grammatical Errors:

4. Line 59: The previous sentence that focuses on influenza makes clear that the authors' inference about airborne transmission stems from data resulting from measurement of airborne samples. However, for RSV (starting at Line 59), none of the various sample locations described refer to airborne samples; therefore, in the next sentence ("Only a small number....") it is not clear whether the authors are arguing that airborne transmission is unlikely due to the scarcity of *airborne* samples or whether airborne transmission is unlikely due to the scarcity of *any* samples. Please clarify.

RESPONSE: Thank you for your insight, we agree that it was confusing. We have modified the sentences to make it clear that we are talking about air samples. Here is the new version: “RSV genomes have been detected in the air of an emergency clinic (20), in hospital rooms with RSV-infected patients (21) and in a pediatric acute care ward (22). Only a small number of positive samples (2.3%) were collected in the air of the pediatric acute care ward, which suggests that airborne RSV transmission is not likely (22).”

5. Line 63: Manuscript would be improved if additional context were given to the characterization of "high" used to describe genome copies per cubic meter of air. Clearly, this depends on distance from the source, air changes per hour imposed by ventilation in the room, etc.

RESPONSE: Since “high” is somewhat arbitrary we modified the two sentences to remove this claim. Here is the new version: “On the other hand, another study detected airborne infectious RSV in the air around infants that were hospitalized with bronchiolitis in a pediatric ward and an intensive care. Mean concentrations were 3.71 X 105 PFU/m3 in the pediatric ward and 4.09 X 105 PFU/m3 in the intensive care unit [23].

6. Line 71: Similar critique as above: the characterization of "low concentrations of ozone" would be greatly improved if additional context were provided. There are multiple contexts to consider. There is the context of prior studies of ozonation for sterilization as well as the limits on ozone concentrations in air for protection of human health. The latter is much lower and therefore likely would undercut any unqualified assertion that the ozone concentrations of the present studies are "low".

RESPONSE: We agree that we only refer to the concentrations used by other studies and not human health. The only study we found that inactivated airborne phages was by Tseng and Li, which we have cited in the modified sentence: “The results that were obtained were promising, even when using lower concentrations of ozone than those used by Tseng and Li (2006) for the inactivation of airborne phages (25).

7. Line 120: The sentence beginning "Diffusion dryer tubes...." is missing a verb.

RESPONSE: Thank you for pointing this out. We have added a verb. “Diffusion dryer tubes of different lengths were used to adjust the RH inside the chamber.”

8. Line 133: Dilute (i.e., ppm-level) concentrations of ozone in air are inaccurately described as "ozone" (solely).

RESPONSE: Thank you for your comment. You are right, the ozone generator converts the oxygen present in the air (around 21%) in ozone. Therefore, the “ozone” is not pure ozone, but a mixed gas of approximately 21% of ozone in air. This has been corrected in the text: “Next, aerosols were exposed to air (designated as the reference conditions) or a mixed gas of approximately 21% of ozone in air for 0, 30 or 60 minutes at low or high RHs.” 

9. Line 145: Why was a new ozone monitor introduced?

RESPONSE: The ozone monitor that we previously used could not be connected with fittings and tubing to the aerosol chamber. Therefore, the ozone concentration had to be assessed with another aerosol chamber that had a window, which was necessary for the ozone probe readings. The acquisition of a new ozone probe allowed us to measure the ozone concentrations directly inside the rotative drum because it could be connected with fittings and tubing. 

10. Line 148: The second phrase in this sentence lacks a verb ("....which 20 ppb and 50 ppb, with .....")

RESPONSE: Here is the sentence with the verb “The sensitivity of the sensors is considered to be 1% of their ranges, which are 20 ppb (500-2000 ppb) and 50 ppb (1-5 ppm), with a resolution of 1 ppb and 0.01 ppm, respectively.”

---

## [Editor Report · Decision Letter 1]

27 May 2021

Ozone inactivation of airborne influenza and lack of resistance of respiratory syncytial virus to aerosolization and sampling processes

PONE-D-21-02103R1

Dear Dr. Duchaine,

We’re pleased to inform you that your manuscript has been judged scientifically suitable for publication and will be formally accepted for publication once it meets all outstanding technical requirements.

Kind regards,

Sander Herfst

Academic Editor

PLOS ONE

Additional Editor Comments:

Please explain why a co-author was added to the manuscript (what was their contribution) and what the reason is that a funding agency was removed from the acknowledgements section? Especially the latter is rather unusual.

---

## [Editor Report · Acceptance letter]

2 Jul 2021

PONE-D-21-02103R1 

Ozone inactivation of airborne influenza and lack of resistance of respiratory syncytial virus to aerosolization and sampling processes 

Dear Dr. Duchaine:

I'm pleased to inform you that your manuscript has been deemed suitable for publication in PLOS ONE. Congratulations! Your manuscript is now with our production department. 

Kind regards, 

on behalf of

Dr. Sander Herfst 

Academic Editor

PLOS ONE